# Testing and Evaluation of the Electric Drive System on the Vehicle Level

**Zhiguo Kong [1,2,\*], Wei Zhang [2] and Helin Zhang [2]**

1   China Automotive Technology and Research Center Co., Ltd., Tianjin 300300, China
2   CATARC New Energy Vehicle Test Center (Tianjin) Co., Ltd., Tianjin 300300, China; zhangwei@catarc.ac.cn (W.Z.); zhanghelin@catarc.ac.cn (H.Z.)
\*   Correspondence: kongzhiguo@catarc.ac.cn; Tel.: +86-22-84379666-6834

**Abstract:** In order to obtain the performance of the electric drive system (EDS) on the vehicle level correctly and effectively, the authors carried out research on the testing and evaluation technology. Firstly, the typical control strategy and its influence and limitations regarding the EDS performance were discussed in detail. Secondly, a test system on four-wheel dynamometers with a high-performance data acquisition and analysis system was introduced. A dedicated test was performed. Further analysis of the results was introduced. It is proved that the method introduced here is feasible and effective, which is beneficial to benchmarking and evaluation of the EDS used in electric vehicles.

**Keywords:** electric driving system; electric vehicle; control strategy; vehicle level

## 1. Introduction

With the people's increasing attention to energy security and environmental pollution, automobile electrification has become the trend [1,2]. Electric vehicles can also serve as a mobile platform in vehicle-to-grid and have good overall benefits [3–5]. The electric drive system (EDS) is the core component of electric vehicles [6]. For bench testing of the EDS, many standards have been issued in China [7], where test methods and criteria are provided. However, part of the EDS's performance, especially the system efficiency, is affected by input voltage, cooling conditions, operating points, which means test results are often idealistic and unspecific. Furthermore, for advanced EDS benchmarking, testing cannot proceed without a protocol. Furthermore, a lot of the vehicles in service need to be assessed to determine whether security risks exist.

There are large differences between testing on the vehicle level and that on the bench [8–12]. The main differences are the power supply, cooling conditions and control mode [9–13]. The power supply in bench testing often uses high-power DC. The capacity is sufficient for the load and the supply remains stable throughout the process. The control command is sent to the motor control unit (MCU) directly through the CAN bus. However, in vehicle-level testing, the power supply is the on-board battery system, and its voltage varies and the current is limited by pack performance. The output of the EDS is controlled by the acceleration pedal and the brake pedal through the vehicle controller. In addition, the performance of the cooling system in vehicle-level testing can affect test results to some extent. In this paper, the typical control strategy and its influence on the EDS performance are discussed. Further, some test criteria are given. Finally, some results are shown and discussed to verify the introduced test method.

## 2. Influence of System Matching and the Control Strategy

In electric vehicles, the output of the EDS is determined by the control strategy [14]. The decisive role is suitable not only for hybrid and fuel-cell vehicles that have a variety of energy sources [15–18], but also for electric vehicles where batteries are the only on-board energy source. For example, the control system determines when and how the

maximum power outputs to the wheels. Generally, the control unit processes the input parameters and output commands with some strategy and rules, which are needed to consider and coordinate the driver's power requirements and the boundary conditions of each subsystem safely and smoothly. The maximum power of the EDS is limited by the maximum allowable charge and discharge power of the on-board energy power supply, especially at low temperature and low SOC. The control method and the measurement of the working point of regenerative braking also need to be considered. To obtain the MAP of the EDS, working conditions should be set carefully.

### 2.1. Typical Control Strategy Analysis

When the EDS drives the vehicle forward, the output power is related to the APO (accelerator pedal opening). Figure 1 shows a typical relationship curve between the output power and the accelerator pedal. It should be noted that the actual output power of the EDS $P_{out}$ is limited not only by its own capability $P_{mot}$ at that speed, but also by the maximum allowable discharge power $P_{bat\_dis}$ of the battery pack in the current state updated by the battery management system (BMS) in real time, i.e., $P_{out} = min(P_{req}, P_{mot}, P_{bat\_dis})$, where $P_{req}$ is the power requirement corresponding to the current acceleration pedal opening as shown in Figure 1. Figure 2 shows a typical curve of the relationship between the output power ratio and the APO. Here, $P_{bat\_dis}$ was determined by the SOC (state of charge) and the battery temperature, and a gain was introduced in view of the efficiency of the EDS. In the reverse mode, the output power was limited with saturation.

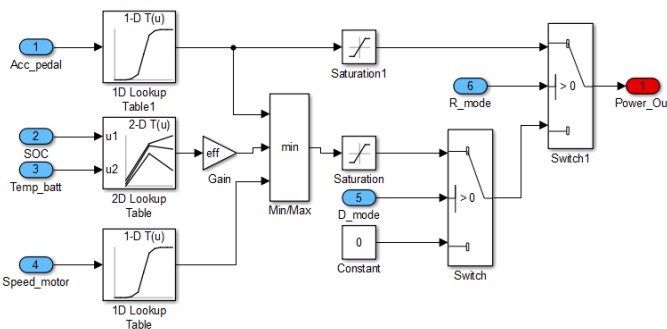

**Figure 1.** Schematic diagram of the actual output power control method.

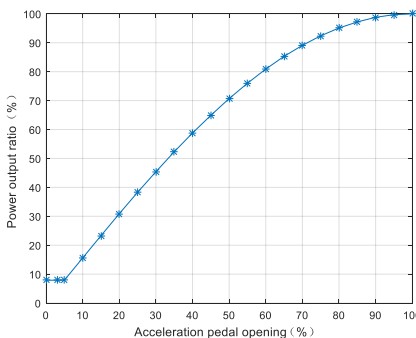

**Figure 2.** Relationship curve between the output power ratio and the APO.

In addition, the performance of the battery pack limits the power of the EDS. Generally, power batteries can produce large peak power for a short time without significantly affecting their life. Some automobile makers allow a short period of high-power output to provide short-term acceleration performance, such as in the 0–100 km/h acceleration testing. However, the trigger conditions are stringent and cannot be used frequently in view of the safety and lifetime of the battery pack. Thus, when we tested the output characteristics of the EDS, we could not detect the nominal peak power and torque and therefore could not obtain the complete MAP of the EDS.

The power generation operation of the motor occurs in the regenerative braking phase when the vehicle is slowed down or braked, which is the main difference between the electric vehicles and the conventional fuel vehicles. Here, the motor works as a generator to convert the vehicle's momentum into electrical energy and store it in a power battery. There are many ways to control the braking process, and the commonly used braking strategies were presented by Junzhi Zhang [11]. Figure 3 shows a diagram of the relationship between the braking power and the brake pedal opening in a common coordinated regenerative brake.

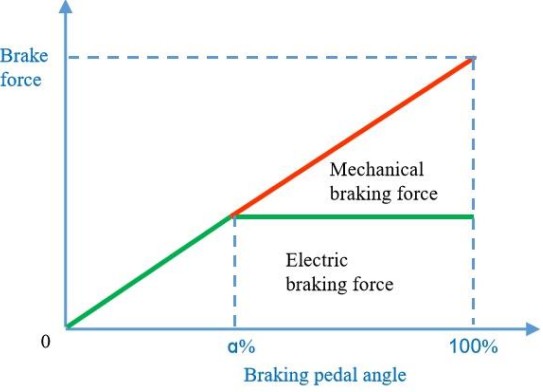

**Figure 3.** Relationship between the braking force and the brake pedal angle.

For safety reasons, the vehicle control system is generally equipped with the brake override function. When the brake pedal is pressed, the vehicle will immediately slow down regardless of whether the accelerator pedal is pressed. Besides, as shown in Figure 1, when the accelerator pedal and the braking pedal are fully released, there is a small power output at low speed to realize the creep control of the vehicle. When the vehicle speed is higher than a certain preset value, the EDS runs as a generator to produce the braking torque to control vehicle sliding. The torque in creeping and gliding often needs to be adjusted and optimized for different vehicle parameters and strategies.

For harshness, the increment adjustment control method has often been used [19,20]. The increment of the torque can be limited to step by step. The upper limit of Saturation4 in Figure 4 determines the growth rate of torque, and the lower limit of Saturation4 determines the deceleration rate of torque, which can be set separately. Furthermore, if you want to change the torque increment at different speeds to meet the corresponding requirements in different speed segments, more increment adjustment parameters can be introduced, such as Saturation2 and Saturation3 in Figure 4. As a result, for large changes in torque demand, the actual torque output just follows the change rather than responds indiscriminately all the time. Only when the demanded torque is stable for a period, the actual output torque can match the instruction. The actual torque output follows the intent of the driver smoothly [20].

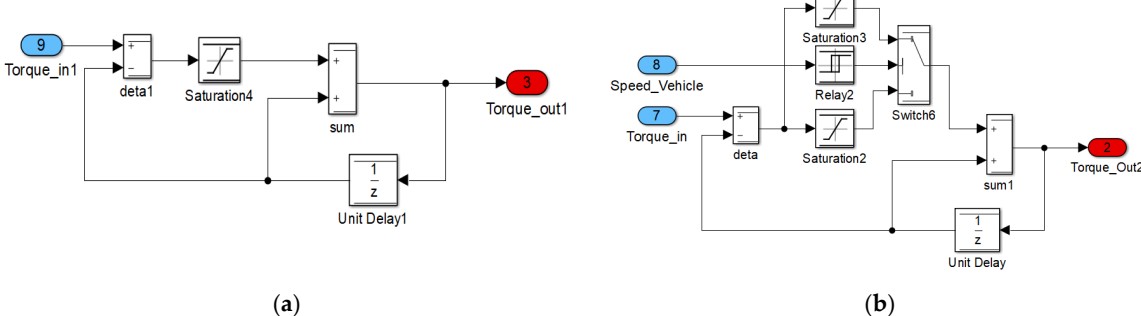

(**a**)                                            (**b**)

**Figure 4.** Schematic diagram of the torque increment adjustment method: (**a**) single rate; (**b**) multi-rate.

*2.2. Impact on Electric Driving System Testing on the Vehicle Level*

Vehicle-level EDS testing is generally selected in both the driving mode and the forward braking mode. Based on the analysis above, the impact of vehicle-level electric drive testing is mainly reflected in the following aspects:

1.  Input voltage control. The allowed voltage range should be selected before the test, and each test point should work as quickly as possible. Once beyond that range, the battery should be charged in time.
2.  Cooling system control. During the test, we should make the vehicle cooling system work properly and maintain the temperature range changes in a specified range. If necessary, we should stop the test and wait a minute until each subsystem has recovered from overtemperature.
3.  Working point control. The test can only control the vehicle output through the pedal opening profile, and thus steady-state testing should maintain a stable opening degree. A dynamic test is needed to adjust the pedal to follow the vehicle's V–T curve (speed–time).
4.  Working point recording. Due to the inevitable voltage changes during testing, it is necessary to collect and analyze the high-speed number simultaneously to obtain the transient power and efficiency of each working point.
5.  Small torque measurement. For the minimum power required in creeping as shown in Figure 2, it is difficult to measure small torque at low speed. The regenerative braking torque during the coast-down testing determined the minimum braking torque in other speed ranges.
6.  Steady-state efficiency calculation. Because of the increment optimization control of the output torque, data recording should be performed after a stable period. For highly synchronous data acquisition systems, this problem can be overcome even for transient recording.

## 3. Performance Evaluation of a Full-Vehicle Electric Drive System

Figure 5 shows the schematic of the test bench. Firstly, four wheels of the vehicle under testing were removed. Then, each output end of the vehicle was connected to a dynamometer through a connecting shaft. All the four dynamometers and the autopilot were controlled by the host computer.

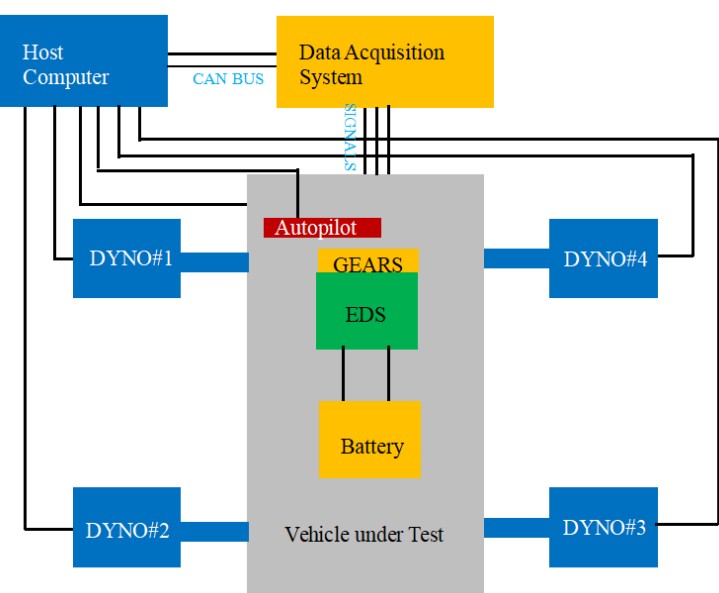

**Figure 5.** Schematic diagram of the test bench on the vehicle level.

Figure 6 shows a picture of the test bench. Some parameters are given in Table 1. The EDS speed was calculated using the speed of dynamometers, and the torque was measured with a wireless telemetry system. The voltage, current and temperature signals could be obtained by respective sensors, as shown in Figure 6. In order to reduce the influence of the measurement error on cycle testing results [21], a high-speed multiaccess synchronous data acquisition system was adopted to collect all the necessary signals. The autopilot adjusted the APO and the BPO (braking pedal opening) during each measurement.

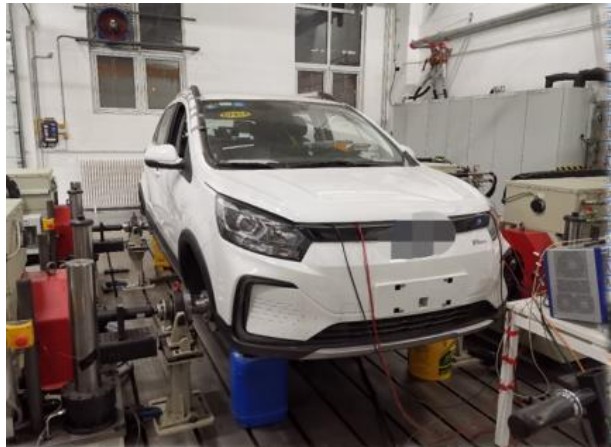

**Figure 6.** Picture of the 4-DYNO test bench.

**Table 1.** Some parameters of the electric vehicle under test.

| Items | Parameters | Items | Parameters |
|---|---|---|---|
| Vehicle weight (kg) | 2300 | Driving mileage (km) | 560 (NEDC) |
| Size (L × W × H (mm)) | 4870 × 1950 × 1725 | Motor type | PM |
| Maximum speed (km/h) | 180 | Driving type | FWD |

### 3.1. Efficiency Testing under Different Cycling Conditions

Different from individual testing, vehicle-level EDS testing pays more attention to the efficiency under different operating cycles rather than the steady-state efficiency [22,23]. This is because the economy and emission performance of the vehicles are tested in such standard operating conditions. The comprehensive efficiency under different working conditions has a more direct influence on the driving range and energy consumption of the vehicle, which is a good reference and can clearly guide design selection and optimization. The most commonly used operating cycling conditions for light vehicles are NEDC, WLTC and CLTC-P [23,24], which are adopted for light vehicle economy testing in several national standards. As an example, the continuous cumulative results of the mechanical energy and the electrical energy changed with time in WLTC cycle testing are provided in Figure 7. The two types of energy were calculated in the driving mode and the braking mode, respectively. Thus, we can get the efficiency in the two modes separately, as shown in Table 2. Before the test, it is necessary to accurately set the resistance on the dynamometers according to the resistance data of the tested vehicle in the test site. Due to the higher average speed and total mileage in the WLTC condition [24], the energy conversion required by the EDS is the most among the three standard cycles, while the energy conversion required in the NEDC condition is the least.

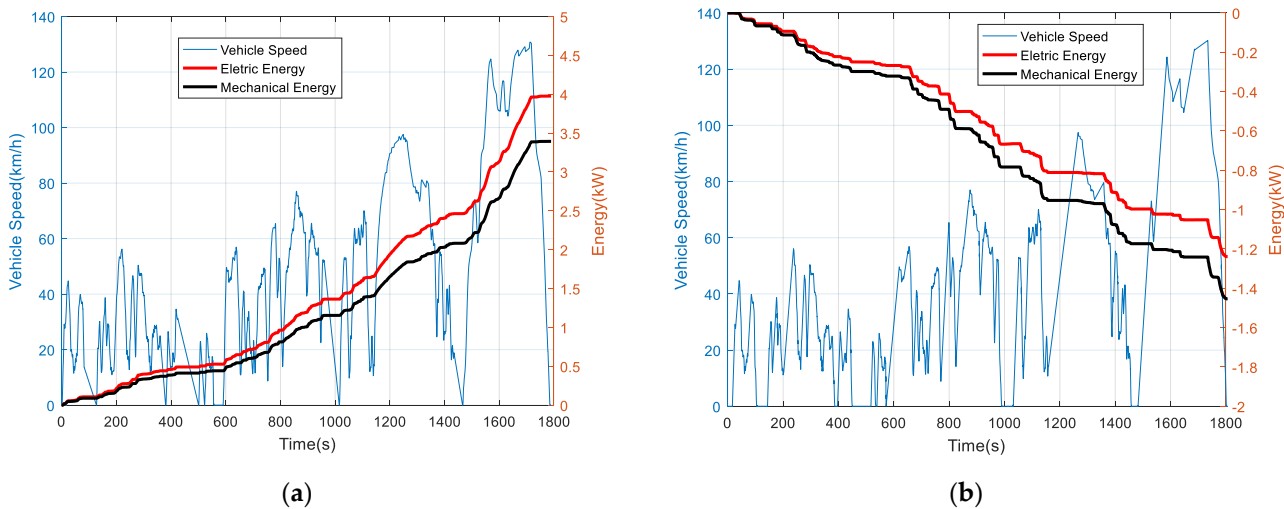

**Figure 7.** Curves of the mechanical and electrical energy changed with time in the WLTC cycle: (**a**) driving; (**b**) regenerative braking.

**Table 2.** Energy and efficiency in each cycle.

| Cycles | NEDC | | WLTC | | CLTC-P | |
|---|---|---|---|---|---|---|
| Modes | Driving | Braking | Driving | Braking | Driving | Braking |
| Electric energy (kWh) | 1.645 | 0.528 | 3.976 | 1.239 | 2.413 | 0.948 |
| Mechanical energy (kWh) | 1.376 | 0.625 | 3.393 | 1.454 | 2.077 | 1.166 |
| Efficiency (%) | 83.65 | 84.48 | 85.34 | 85.20 | 86.07 | 81.30 |

### 3.2. Distribution of Working Points under Different Cycles

In the EDS working test under the three cycles mentioned above, the data sampling frequency was 10 Hz. The EDS's working points can be dotted to obtain the distribution under the three working conditions above, as shown in Figure 8. Firstly, the working points of the EDS were concentrated in the low-torque region at each speed. Secondly, due to the design of the maximum vehicle speed and the speed distribution of typical cycles, the distribution of working points was concentrated in the middle- and low-speed regions. Besides, the operating efficiency of the power generation system should focus on the low-torque area. The working point distribution region should be the key point in the design and optimization of the EDS.

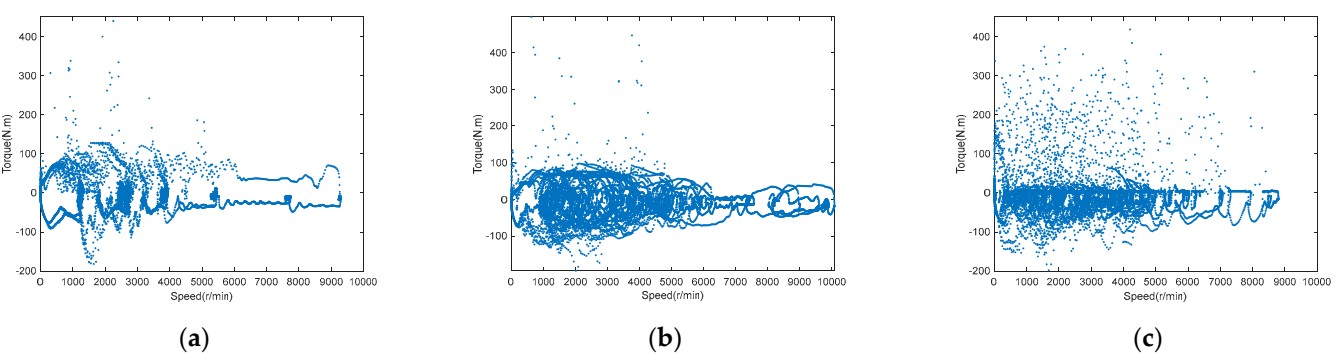

**Figure 8.** Distribution of the EDS under three typical cycles: (**a**) NEDC; (**b**) WLTC; (**c**) CLTC-P.

### 3.3. External Characteristics Test

As mentioned above, we can obtain many measuring points by adjusting different APO and BPO. However, the most concerned curve in vehicle-level EDS measurement [25] is the output characteristic curve with 100% APO, as shown in Figure 9a. Here, the torque and speed of the EDS were measured on the output terminals of the reducer. The EDS

could output the peak power and torque within the allowable SOC operating range of the vehicle [26]. Figure 9b shows the output torques with speed in the SOC = 98.8%, 56% and 21%. There are no obvious differences in the speed range, which shows good output consistency. Since the maximum allowable output power of the battery remains constant in a certain range of the SOC and the SOH [27,28], the output power of the power system remains almost the same. The test result was in accordance with the expectation.

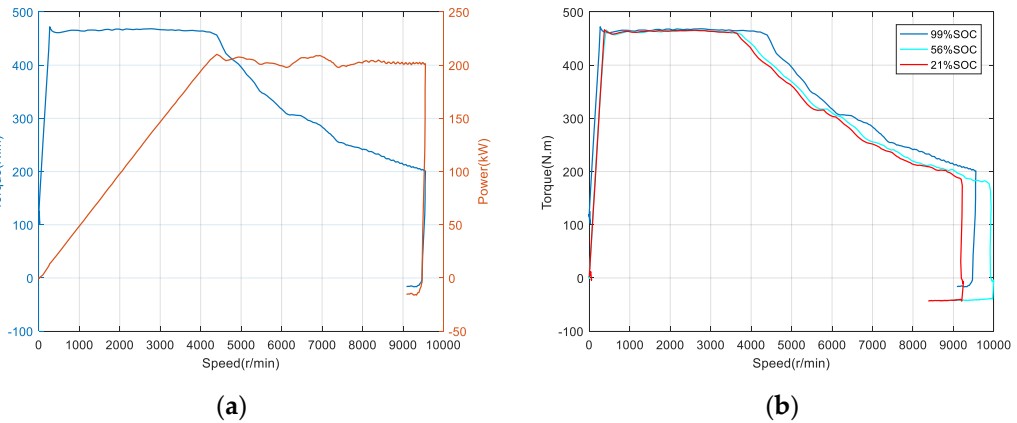

| (a) | (b) |

**Figure 9.** Curves of the output torque and power in (**a**) APO = 100%; (**b**) different SOC.

### 3.4. EDS's Temperature Rise Test

The temperature rise performance of the EDS relates to the reliability and security of the vehicle. It is necessary to examine EDS's temperature rise in typical scenarios, such as daily driving, long periods of constant speed driving, sharp acceleration and long periods of uphill driving. The temperature variation of the EDS in the cycles indicated that the temperature rise of the EDS could meet the requirements of vehicle operation during daily driving and sharp acceleration. Furthermore, the vehicle could run at a constant speed for a long time with reasonable working point selection of the EDS and the cooling capacity. In vehicle-level testing, the temperature rise of the EDS should be paid more attention to in case of a long climb. In the measurement, the vehicle was set to proceed at 50 km/h, which corresponds to the rated speed of the EDS. The accelerator pedal was pushed fully, APO = 100%. The curves of the output power of the EDS and the motor temperature are shown in Figure 10. It took about 114 s for the temperature to rise from 54 °C to 140 °C, and then the output power was limited to avoid further temperature rise, which would cause performance deterioration. With the APO kept at 100%, the output power dropped to zero and the temperature reaching 150 °C, the test time was approximately 123 s. Temperature rise can meet the needs of most scenarios. For the sake of safety, it is recommended to reduce the vehicle speed and take regular breaks in case of a long climb.

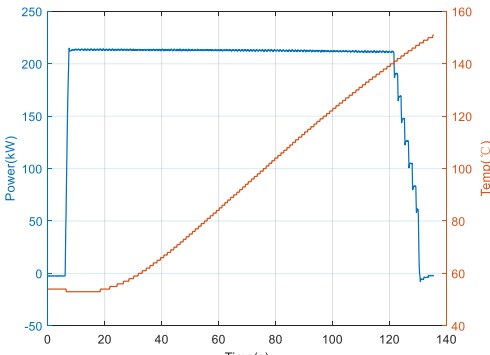

**Figure 10.** Curves of the output power and the motor temperature with time.

## 4. Discussion

This paper presented the vehicle-level EDS measurement technology after the control strategy analysis. The steady-state point efficiency test is affected by the performance of the vehicle control strategy and the battery pack. It is difficult to obtain the full-plane characteristic data of the EDS on the vehicle level. Comprehensive and detailed characteristics of the EDS still come from bench test results; this requires an accurate and open communication protocol support.

Considering the requirement of system design and vehicle application, four aspects of testing were recommended and discussed in detail, including cycling conditions efficiency, working points distribution, external characteristics and temperature rise. The analysis of the test results can provide direct and accurate guidance for vehicle matching design. Based on the test results of the actual component characteristics and the vehicle environmental boundary conditions, the performance of the vehicle power system can be optimized further.

Furthermore, vehicle-level EDS testing is necessary and feasible in consideration of more and more advanced EDS benchmarking and a large number of vehicle performance evaluations in use.

**Author Contributions:** Conceptualization, Z.K.; validation, H.Z.; data curation, W.Z.; writing—original draft preparation, Z.K.; writing—review and editing, Z.K.; visualization, Z.K. All authors have read and agreed to the published version of the manuscript.

**Funding:** This research received no external funding.

**Conflicts of Interest:** Zhiguo Kong is an employee of China Automotive Technology and Research Center Co., Ltd. and CATARC New Energy vehicle test center (Tianjin) Co., Ltd. Wei Zhang and Helin Zhang are employees of CATARC New Energy vehicle test center (Tianjin) Co., Ltd. The paper reflects the views of the scientists, and not the company.

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
