# Peer review of "Testing and Evaluation of the Electric Drive System on the Vehicle Level"

_wevj, doi:10.3390/wevj12040182_

Round 1
Reviewer 1 Report
- The literature review is poor. Reference list consists only 9 references.
- The research gap should be outlined properly in the introduction. It is desirable to broaden the rationale for the study.
- Graphics can be improved.
Author Response
1. literature review has been extend from 9 to 28 in difference pars, as shown in
Reviewer 2 Report
The problem presented in the article "Test and Evaluation of Electric Drive System in Vehicle Level " is very interesting. But, the article requires a major revision.
- What is the control and optimization strategy?
- What were the parameters of the tested vehicle.
- Figure 8 shows the NEDC, WLTC, CLTC-P tests with a rotational speed of up to 10,000 RPM.Why the presented test results in Fig. 9 are up to the speed of 1000 RPM.
- Why are the tests shown in Fig. 10 carried out at a temperature of 54 degrees Celsius?
- Please correct the figures 1, 2, 7, 8, 9.
- In line 97, Fig. 4 is mentioned, but not shown.
- What does MAP mean.
- The conclusions are very modest in relation to the research conducted.They should be extended.
- Extend the literature e.g. with items related to SOC and NEDC:
- Cubito C., Millo F., Boccardo G., Di Pierro G., Ciuffo B., Fontaras G., Serra S., Garcia M. O., Trentadue G.: Impact of Different Driving Cycles and Operating Conditions on CO2 Emissions and Energy Management Strategies of a Euro-6 Hybrid Electric Vehicle, Energies 2017, 10, 1590; doi:10.3390/en10101590
- Tribioli, L. Energy-Based Design of Powertrain for a Re-Engineered Post-Transmission Hybrid Electric Vehicle. Energies 2017, 10, 918, org/10.3390/en10070918
Overall, this paper is interesting, but needs to be revised.
Author Response
- This paper only shows a common control strategies to analyze the impact on test method and results, and there is no optimization.
- This parameter of the vehicle is given in revised version.
- The motor output with reducer, and it was tested in assembly form. The motor speed was given on CAN bus, and the speed in fig.9 are the powertrain, I have corret it to be sonsistent.
- The 54℃ is the the temperature after the last test, we evaluated that the temperature rise test could be completed from now on as shown in fig.10, and we did not specifically select the starting temperature point.
- We have corrent figures 1,2, 7,8,9.
- We have added figrues 4. Thanks a lot.
- The efficiency contour drawn with the rotational speed as the horizontal axis and the torque as the vertical axis is called the motor MAP, which is a common term.
- We have extended the conclusion according to the research.
- We have added some literatures from 9 to 28, including items related to SOC and NEDC. Thanks for your kindly recommendation

Round 2
Reviewer 1 Report
Revised manuscript can be accepted for publication.
Reviewer 2 Report
Hi,
I suggest the Authors change the description of the drawings:
Figure 4. Schematic diagram of the torque increment adjustment method (a) Single rate (b) Multi-rate.
Figure 7. Curves of mechanical and electrical energy changed with time in WLTC cycle (a) Driving (b) Regenerative braking.
Regards
Reviewer